# The Association Between Glucose Variability and Insulin Parameters in Gestational Diabetes Diagnosed After 24 Gestational Weeks

**DOI:** 10.3390/nu17030440

**Published:** 2025-01-25

**Authors:** Yoshifumi Kasuga, Kaoru Kajikawa, Naotsugu Ishikawa, Yasuhiko Ogata, Marina Takahashi, Keisuke Akita, Junko Tamai, Yuka Fukuma, Yuya Tanaka, Toshimitsu Otani, Marie Fukutake, Satoru Ikenoue, Mamoru Tanaka

**Affiliations:** Department of Obstetrics and Gynecology, Keio University School of Medicine, 5 Shinanomachi, Shinjuku-ku, Tokyo 160-8582, Japan

**Keywords:** gestational diabetes mellitus, glucose variability, insulin, oral glucose tolerance test, insulinogenic index, homeostasis model assessment—insulin resistance, whole-body insulin sensitivity index derived from the OGTT, Japanese

## Abstract

**Background/Objectives**: Recently, it was reported that glucose variability (GV) calculated using the 75 g oral glucose tolerance test (OGTT) is associated with adverse perinatal outcomes. However, its role in gestational diabetes mellitus (GDM) remains unclear. We investigated the association between GV and insulin parameters in Japanese women diagnosed with GDM after 24 weeks of gestation (late GDM). **Methods**: A total of 280 mothers with late GDM cared for at Keio University Hospital were included in this study. Using 75 g OGTT, the initial increase and subsequent decrease were calculated as the GV. **Results**: The initial increase was significantly positively associated with 1 h plasma glucose level (PG) and 2 h PG with 75 g OGTT (*p* < 0.001), but fasting PG, insulinogenic index (IGI), and homeostasis model assessment—insulin resistance were negatively associated with the initial increase (all *p* < 0.001). The subsequent decrease was significantly positively correlated with 1 h PG (*p* < 0.001) but negatively correlated with 2 h PG (*p* < 0.001), IGI (*p* = 0.009), and the whole-body insulin sensitivity index derived from the OGTT (*p* = 0.02). Insulin Secretion-Sensitivity Index-2 was not associated with an initial increase or subsequent decrease. **Conclusions**: Since the initial increase might reflect insulin secretion and the subsequent decrease might reflect insulin sensitivity in Japanese women with late GDM, GV could alter several insulin parameters. Further studies are required to investigate the usefulness of GV in the management of GDM.

## 1. Introduction

Gestational diabetes (GDM) is one of the most common perinatal complications, and the incidence of GDM in Japanese women is approximately 10%. GDM is associated with adverse perinatal outcomes (e.g., fetal overgrowth, neonatal hypoglycemia, and cesarean section delivery) [1,2,3,4]. To predict adverse outcomes in women with GDM, it is important to evaluate certain parameters during pregnancy. For example, since neonatal hypoglycemia may predict obesity, metabolic syndrome, and neurodevelopment in the future, this could pose a very important challenge for the healthcare of the next generation [5,6,7]. In our previous DNA methylation analysis of GDM using Japanese umbilical cord blood samples, while GDM did not change DNA methylations compared to normal glucose tolerance [8], the promoter regions of *ZNF696,* which are correlated with glucose tolerance, were associated with neonatal hypoglycemia [9]. Glycated hemoglobin (HbA1c) levels and insulin therapy during pregnancy may be good predictors of neonatal hypoglycemia [10,11,12]. Recently, glucose variability (GV), calculated using the 75 g oral glucose tolerance test (75 g-OGTT), has been reported as a predictor of adverse perinatal outcomes. Tano et al. reported that GV at early gestation is a good predictor of hypertensive disorders of pregnancy (HDP) [13]. In our previous report on GDM diagnosed before 24 gestational weeks (early GDM), both the initial increase and subsequent decrease in the insulin therapy group before 24 gestational weeks were higher than those in the insulin therapy group after 24 gestational weeks and diet therapy group [14].

Several clinicians have evaluated insulin secretion and resistance to GDM. In Caucasians, insulin resistance is involved in GDM [15]. On the other hand, in our previous studies, impaired insulin secretion and beta cell dysfunction played a role in the development of GDM in Japanese mothers based on both clinical and genetic data [16]. Measuring insulin parameters is crucial for understanding the development of GDM. However, without measuring plasma insulin levels, it is impossible to know insulin parameters. Furthermore, measuring plasma insulin levels with 75 g OGTT is not common practice in clinical settings in Japan. Although GV might be important for evaluating GDM outcomes and is easy to calculate using 75 g OGTT, its association with insulin parameters in mothers with GDM has not been investigated.

Therefore, we aimed to investigate the association between GV and insulin parameters in Japanese women diagnosed with GDM after 24 gestational weeks (late GDM).

## 2. Materials and Methods

This study enrolled mothers diagnosed with GDM after 24 weeks of gestation who were cared for at Keio University Hospital between January 2017 and December 2019 (*n* = 280). Mothers with GDM were diagnosed using 75 g OGTT according to the Japan Society of Obstetrics and Gynecology (JSOG) diagnostic criteria modified by the International Association of Diabetes and Pregnancy Study Group [17]. Therefore, the cut-off values of 75g OGTT were fasting plasma glucose level (FPG) ≥ 92 mg/dL, 1 h plasma glucose level (1 h PG) ≥ 180 mg/dL, and 2 h plasma glucose level (2 h PG) ≥ 153 mg/dL. The management of GDM at our hospital has been described in detail in our previous report [18]. All mothers with GDM performed self-blood glucose monitoring before and 2 h after all planned meals and at bedtime in this study. Diet therapy was started immediately after they were diagnosed with GDM. The number of calories for diet therapy was decided based on maternal height and pre-pregnancy body mass index (BMI), height (m)^2^ × 22 × 30 kcal (pre-pregnancy BMI ≥ 25), and height (m)^2^ × 22 × 30 + 350 kcal (pre-pregnancy BMI < 15). When their PG before a meal was >90 mg/dL or that at 2 h after a meal was 120 mg/dL, the mothers were administered insulin therapy. We excluded mothers with GDM with multi-fetal pregnancies, those diagnosed before 24 gestational weeks, and those lacking insulin parameters. This study was reviewed and approved by the Keio University Hospital Ethics Committee (approval nos. 20150103 and 20150168) and conformed to the provisions of the Declaration of Helsinki. Since this was a retrospective study, the need for informed consent from patients was waived.

Using 75 g OGTT, the initial increase was calculated as 1 h PG–FPG (mg/dL) and the subsequent decrease was calculated as 1 h PG–2 h PG (mg/dL) [13]. We also measured PG at 30 min (PG_30_) and plasma insulin levels at 0 min (Ins_0_), 30 min (Ins_30_), 60 min (Ins_60_), and 120 min (Ins_120_) with 75 g OGTT. We evaluated the insulinogenic index (IGI), whole-body insulin sensitivity index derived from the OGTT (IS_OGTT_), homeostasis model assessment of insulin resistance (HOMA-R), and Insulin Secretion-Sensitivity Index-2 (ISSI-2). The IGI was calculated as ({Ins_30_ − Ins_0_}/{PG_30_ − FPG}) and the ratio of the total area under the insulin curve to the total area under the glucose curve (AUC_ins/glu_) with the OGTT [19]. The IS_OGTT_ was calculated using the following formula: 10,000/square root {FPG × Ins_0_ × (FPG + 1 h PG × 2 + 2 h PG)/2 × (Ins_0_ + Ins_60_ × 2 + Ins_120_)/2}, where PG_y_ (mg/dL) and Ins_y_ (mU/L) represent plasma glucose and insulin levels, respectively, at time y min with the OGTT [20]. HOMA-R was calculated as follows: Ins_0_ × FPG/405 [21]. ISSI-2 was calculated as AUC_ins/glu_ multiplied by IS_OGTT_ [22]. Neonatal hypoglycemia was defined as PG < 47 mg/dL at 1, 2, or 4 h postpartum [10].

Comparisons of maternal and perinatal outcomes between the insulin and diet therapy groups were performed using the chi-square test or Fisher’s exact test for categorical variables and the Mann−Whitney U-test for continuous variables. The trend in the number of abnormal PG with 75 g OGTT was evaluated using Cochran–Armitage trend analysis. Since the Shapiro–Wilk test did not show that the data of GV and insulin parameters had a consistent distribution, the correlation between GV and insulin parameters was calculated using Spearman’s test. To investigate the risks of insulin therapy, insulin parameters and GV were evaluated using multiple logistic regression analysis, adjusted for maternal age at delivery and pre-pregnancy BMI. *p* < 0.05 indicated statistical significance. Statistical analysis was performed using JMP software (ver.17, SAS Institute, Cary, NC, USA) and R (ver. 4.3.2, https://cran.r-project.org/)(accessed on 1 November, 2024).

## 3. Results

A comparison of the maternal characteristics and perinatal outcomes between the insulin and diet therapy groups is shown in Table 1. The incidences of nulliparity, family history of diabetes, abnormal FPG, abnormal 1 h PG, abnormal 2 h PG, cesarean section (CS), and small for gestational age (SGA; birth weight < 10th percentile) were significantly higher in the insulin therapy group than in the diet therapy group. The incidence of elective CS was similar between the two groups. FPG, 1 h PG, 2 h PG, number of abnormal OGTT, HOMA-R, initial increase, and umbilical cord artery pH in the insulin therapy group were significantly higher than those in the diet therapy group. However, gestational weight gain, IS_OGTT_, ISSI-2, gestational weeks at delivery, and birth weight were significantly lower than those in the diet therapy group. There were no differences in IGI and subsequent decreases between the two groups.

The associations between the insulin parameters and GV in the insulin therapy group are shown in Table 2. According to the multiple logistic regression analysis, HOMA-R, IS_OGTT_, ISSI-2, initial increase, and subsequent decrease were significantly correlated with insulin therapy in Japanese women with late GDM but not IGI.

The association between GV and insulin parameters is shown in Figure 1. The initial increase was significantly positively associated with 1 h PG and 2 h PG (*p* < 0.001); however, FPG, IGI, and HOMA-R were significantly negatively associated with the initial increase (*p* < 0.001). The subsequent decrease was significantly positively correlated with 1 h PG (*p* < 0.001) but negatively correlated with 2 h PG (*p* < 0.001), IGI (*p* = 0.009), and IS_OGTT_ (*p* = 0.02). The ISSI-2 was not associated with an initial increase or subsequent decrease.

Among the objectives, neonatal hypoglycemia was identified in 77 neonates; however, the results of neonatal PG could not be obtained from the medical records of 66 neonates. The initial increase did not differ between the neonatal hypoglycemia (median: 86 mg/dL, range: 23–157 mg/dL) and non-hypoglycemia (median: 93 mg/dL, range: 24–156 mg/dL) (*p* = 0.13) groups. However, the subsequent decrease was significantly lower in the neonatal hypoglycemia group (median: 12 mg/dL, range: −67 to 117 mg/dL) than in the non-hypoglycemia group (median: 18 mg/dL, range: −63 to 108 mg/dL) (*p* < 0.05).

## 4. Discussion

Among mothers with GDM, the initial increase in the insulin therapy group was significantly higher than that in the diet therapy group. However, the subsequent decrease was similar between the two groups. Analysis of the association between GV and insulin parameters showed that the initial increase was significantly negatively correlated with IGI and HOMA-R. A subsequent decrease was significantly negatively correlated with IGI and IS_OGTT_. However, ISSI-2 expression was not associated with GV.

GV is reportedly a good predictor of adverse outcomes in patients with impaired glucose tolerance and is calculated using continuous glucose monitoring (CGM) [23,24]. In non-pregnant patients with diabetes, increased GV is associated with adverse outcomes (e.g., mortality in the intensive care unit and hypoglycemia) [23]. In pregnant women with type 1 diabetes, increased GV was associated with neonatal hypoglycemia [25], and fetal growth was significantly positively correlated with GV in pregnant women with pre-diabetes [26]. However, it is challenging to manage all mothers with GDM using CGM in Japan, and there is a paucity of data regarding GV in GDM. Recently, it was reported that GV could be calculated using 75 g OGTT when CGM cannot be used [27] and that GV calculated using 75 g OGTT can predict HDP [13]. Furthermore, initial increase and subsequent decrease could be predicted in the insulin therapy group before 24 gestational weeks in early GDM [14]. In this study, since an initial increase and subsequent decrease were correlated with the risk of insulin therapy among patients with late GDM, GV might be a good predictor of insulin therapy. Therefore, GV calculated using 75 g OGTT might have the potential to predict adverse outcomes in GDM.

Since the association between GV and insulin parameters has been unknown, we aimed to evaluate the association in this study. IGI refers to the initial response of insulin secretion to oral glucose [19]. Patients with GDM have lower IGI than normal glucose tolerance (NGT) [28]. Lower IGI was associated with impaired postpartum glucose tolerance in Japanese women with prior GDM [29,30,31]. Therefore, IGI may predict adverse outcomes related to impaired insulin secretion. HOMA-R was used to assess insulin resistance. Compared with NGT, patients with GDM had higher HOMA-R [16], and HOMA-R in GDM was associated with large for gestational age (birthweight ≥ 90th percentile) in Japanese neonates [32]. However, because Japanese individuals have impaired insulin secretion and the incidence of obesity is not as high as that in other developed countries, Inoue et al. reported that insulin resistance is not associated with the development of GDM [33]. As IS_OGTT_ is a parameter of insulin sensitivity, the IS_OGTT_ in the high and middle adipose tissue insulin resistance (ATIR) groups was significantly lower than that in the low ATIR group in young, healthy, non-pregnant Japanese women [34]. Pre-pregnancy overweight and obesity were associated with an impaired IS_OGTT_, and IS_OGTT_ was negatively correlated with birth weight, whereas there was no association with polycystic ovary syndrome [35]. Therefore, based on our results, the initial increase may indicate insulin secretion and a subsequent decrease may indicate insulin sensitivity. ISSI-2 assesses beta-cell function. In this study, because the initial increase and subsequent decrease were not associated with ISSI-2, the participants might not have been aware of beta-cell function. Therefore, GV may be a useful tool for managing GDM because it can change several insulin parameters.

This study has some limitations. First, it was a retrospective study with a small sample size. However, to our knowledge, this is the first study to evaluate the correlation between GV using 75 g OGTT and insulin parameters in GDM. Furthermore, since all mothers were cared for in our hospital, they received the same management. Second, according to the JSOG criteria, pregnant women have two chances of being diagnosed with GDM during pregnancy [36,37]. In the first trimester, all pregnant women undergo screening for early GDM, and they undergo 75 g OGTT immediately if they have positive screening results [18]. Obesity was included as a screening criterion. Therefore, the incidence of obesity in early GDM is higher than that in late GDM [18]. In the second trimester, all mothers without early GDM receive a 50 g glucose challenge test or random PG, and they receive 75 g OGTT immediately if they have positive results [18]. Therefore, if the association between GV and insulin parameters is evaluated in the context of early-stage GDM, the results may change. Third, we could not assess other adverse outcomes, such as LGA and macrosomia, as they were not included in this study. Although the subsequent decrease was significantly lower in the neonatal hypoglycemia group than in the non-hypoglycemia group in the present study, we could not obtain the results of neonatal PG from the medical records of over 60 neonates. Therefore, studies with larger sample sizes are needed to evaluate the association between GV and neonatal hypoglycemia. Fourth, as our hospital is a university hospital that primarily cares for high-risk mothers, there might be a selective bias in the present study. Fifth, in the present study, we evaluated GV using 75 g OGTT in Japanese mothers with GDM. However, the results of 75 g OGTT might change after several tests [38,39,40]. Therefore, it might be preferable to use GV measured by CGM when evaluating GV in Japanese mothers with GDM. However, since managing GDM using CGM is not popular in Japan, evaluating GV using CGM was difficult. Further research is required to investigate the usefulness of GV using CGM in Japanese mothers with GDM.

## 5. Conclusions

Since the initial increase might reflect impaired insulin secretion and the subsequent decrease might reflect impaired insulin sensitivity in Japanese women with late GDM, GV could alter several insulin parameters. Further studies are required to investigate the usefulness of GV in the management of GDM and reveal the association between maternal and neonatal long-term outcomes and GV.

## Figures and Tables

**Figure 1 nutrients-17-00440-f001:**
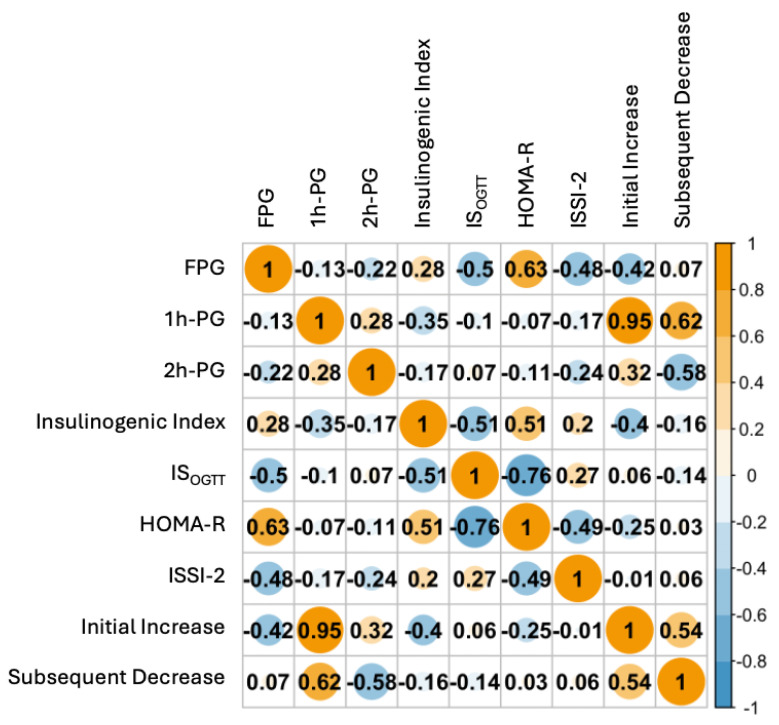
Association between glucose variability and insulin parameters in gestational diabetes mellitus (GDM) diagnosed after 24 gestational weeks. OGTT, oral glucose tolerance test; HOMA-R, homeostasis model assessment of insulin resistance; ISOGTT, whole-body insulin sensitivity index derived from the OGTT; ISSI-2: Insulin Secretion-Sensitivity Index-2.

**Table 1 nutrients-17-00440-t001:** Comparison of maternal characteristics and perinatal outcomes between the insulin therapy and diet therapy groups.

		Insulin Therapy		Diet Therapy		*p*-Value
		(n = 109)		(n = 171)		
Maternal age at delivery	(years)	37	(25−59)	37	(23−51)	0.28
Nulliparity		79	(72%)	116	(68%)	0.001
Pre-pregnancy BMI	(kg/m^2^)	21.1	(16.4−35.4)	20.4	(16.4−34.4)	0.43
Gestational weight gain	(kg/40 weeks)	7	(−7.3−17.8)	9.4	(−5.9−20.8)	0.002
Family history of diabetes		18	(17%)	9	(5%)	0.003
75 g OGTT at diagnosed gestational weeks		27	(24−35)	27	(24–34)	0.01
Fasting glucose level	(mg/dL)	89	(66−138)	86	(68−103)	0.025
1 h glucose level	(mg/dL)	189	(106−250)	172	(103−243)	<0.001
2 h glucose level	(mg/dL)	164	(118−235)	153	(68−230)	<0.001
Abnormal fasting glucose level		38	(35%)	54	(32%)	0.60
Abnormal 1 h glucose level		77	(71%)	59	(35%)	<0.001
Abnormal 2 h glucose level		91	(83%)	104	(61%)	<0.001
Number of positive points						<0.001
1 point positive		27	(25%)	128	(75%)	
2 point positive		67	(61%)	40	(23%)	
3 point positive		15	(14%)	3	(2%)	
HOMA-R		1.59	(0.49−5.11)	1.38	(0.36−6.16)	0.017
IS_OGTT_		4.87	(1.43−13.1)	5.63	(1.69−17.75)	<0.001
ISSI-2		1.59	(0.58−3.52)	1.89	(0.75−4.26)	<0.001
IGI		0.62	(0.16−4.01)	0.69	(0.02−3.42)	0.34
Initial increase	(mg/dL)	101	(23−155)	85	(11−157)	<0.001
Subsequent decrease	(mg/dL)	16	(−67−77)	17	(−63−117)	0.77
Gestational weeks at delivery	(week)	37	(24−41)	38	(28−41)	0.004
Neonatal sex (female)		50	(46%)	78	(46%)	1
Birthweight	(g)	2718	(333−3924)	2962	(670−3974)	<0.001
Cesarean section		64	(59%)	73	(43%)	<0.05
Elective cesarean section		32	(50%)	42	(58%)	0.40
Small for gestational age (birthweight < 10th percentile)		16	(15%)	10	(6%)	0.02
Large for gestational age (birthweight ≥ 90th percentile)		11	(10%)	26	(15%)	0.28
Macrosomia (birthweight ≥ 4000 g)		0	(0%)	0	(0%)	0.13
Apgar score 1 min		8	(1−9)	8	(1−10)	0.66
Apgar score 5 min		9	(2−10)	9	(5−10)	0.59
Umbilical cord artery pH		7.33	(7.10−7.45)	7.31	(7.07−7.48)	0.002

BMI, body mass index; OGTT, oral glucose tolerance test; HOMA-R, homeostasis model assessment of insulin resistance; IS_OGTT_, whole-body insulin sensitivity index derived from the OGTT; ISSI-2: Insulin Secretion-Sensitivity Index-2; IGI: Insulinogenic Index. Data are presented as the median (range: minimum and max) or n (%).

**Table 2 nutrients-17-00440-t002:** Associations between insulin parameters and glucose variability in the insulin therapy group among patients with gestational diabetes mellitus.

Parameter	Single		Multiple	
	OR (95%CI)	*p*-Value	OR (95%CI)	*p*-Value
HOMA-R	1.20 (0.93–1.57)	0.17	0.48 (0.26–0.90)	<0.05
IS_OGTT_	0.82 (0.73–0.93)	0.001	0.68 (0.53–0.86)	<0.001
ISSI-2	0.32 (0.17–0.60)	<0.001	0.18 (0.07–0.48)	<0.001
IGI	1.01 (0.64–1.60)	0.99	2.00 (0.90–4.41)	0.08
Initial increase	1.02 (1.01–1.03)	0.07	1.04 (1.02–1.05)	<0.001
Subsequent decrease	1.00 (0.99–1.00)	0.41	0.98 (0.96–0.99)	<0.001

OR, odds ratio; CI, confidence interval; HOMA-R, homeostasis model assessment of insulin resistance; IS_OGTT_, whole-body insulin sensitivity index derived from the OGTT; ISSI-2: Insulin Secretion-Sensitivity Index-2; IGI: Insulinogenic Index.

## Data Availability

The data presented in this study are available upon request from the corresponding author.

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
