# Peer review of "The Association Between Glucose Variability and Insulin Parameters in Gestational Diabetes Diagnosed After 24 Gestational Weeks"

_nutrients, 2025, doi:10.3390/nu17030440_

Round 1

Reviewer 1 Report

Comments and Suggestions for Authors

In this study the authors aim to investigate the association between glucose variability (GV), measured through OGTT, and insulin parameters in Japanese women diagnosed with GDM after 24 gestational weeks.

I have several concerns.

It is well known that OGTT can have large variability in the same subject. Thus, it is hard to hypotize that a single OGTT may be a good parameter of glucose variability during later gestation.

The results show that GV is able to detect differences in insulin parameters. Nevertheless, these differences have no impact in neonatal outcomes. Thus, the significance of this detection seems unclear.

Were the enrolled women consecutive?

Have women assuming drugs interfering with glucose homeostasis before and during pregnancy been excluded?

How many women performed self-blood glucose monitoring, with fasting and 1 h postmeal capillary measures?

When was medical therapy added to diet and physical activity?

How many women for group underwent cesarean section?

How many cesarean sections were programmed?

How do the authors explain that no macrosomia was observed?

How do the authors explain that rate of newborn large for gestational age was more in diet group than in insulin group?

Despite the enrolled population was largely nulliparous, the age at delivery was high. Do the authors confirm the value of quartiles (25-59 years)?

Author Response

Reviewer 1

In this study the authors aim to investigate the association between glucose variability (GV), measured through OGTT, and insulin parameters in Japanese women diagnosed with GDM after 24 gestational weeks.

I have several concerns. It is well known that OGTT can have large variability in the same subject. Thus, it is hard to hypotize that a single OGTT may be a good parameter of glucose variability during later gestation. The results show that GV is able to detect differences in insulin parameters. Nevertheless, these differences have no impact in neonatal outcomes. Thus, the significance of this detection seems unclear.

Were the enrolled women consecutive?

[Response]

We appreciate this question. Yes, the women were consecutively enrolled.

Have women assuming drugs interfering with glucose homeostasis before and during pregnancy been excluded?

[Response]

We appreciate this question. In Japan, any medications to control plasma glucose levels are not allowed in pregnant women. Thus, the pregnant women enrolled in this study did not use these drugs.

How many women performed self-blood glucose monitoring, with fasting and 1 h postmeal capillary measures?

[Response]

We appreciate this question. As stated in our previous reports, all mothers with GDM performed self-blood glucose monitoring. To address this feedback, we have included the following sentence in the Methods section of the manuscript:

“All mothers with GDM performed self-blood glucose monitoring before and 2 h after all planned meals and bedtime in this study.” (p2, lines 74−75)

When was medical therapy added to diet and physical activity?

[Response]

We appreciate this question. In our hospital, when mothers were diagnosed with GDM, they started diet therapy immediately. However, it is not common to perform physical therapy. Therefore, we have included the following sentence:

“Diet therapy was started immediately after they were diagnosed with GDM.” (p2, lines 75−76)

How many women for group underwent cesarean section? How many cesarean sections were programmed?

[Response]

We appreciate these questions. To address this feedback, we have included the following sentences in the Results and revised Table 1 for clarity:

“The incidences of nulliparity, family history of diabetes, abnormal FPG, abnormal 1h-PG, abnormal 2h-PG, cesarean section (CS), and small for gestational age (SGA; birth weight < 10th percentile) were significantly higher in the insulin therapy group than in the diet therapy group. The incidence of elective CS was similar between the two groups.” (p3, lines 110−114)

How do the authors explain that no macrosomia was observed?

[Response]

We appreciate this question. As described in our previous report, mothers with GDM were well managed at our hospital (Kasuga et al. BMC Open Diabetes Res Care, 2022). Therefore, macrosomia was not observed in this study.

How do the authors explain that rate of newborn large for gestational age was more in diet group than in insulin group?

[Response]

We appreciate this question. Actually, the rate of LGA in the diet therapy group was 15%, and that in the insulin therapy group was 10%. However, this difference was not statistically significant.

Despite the enrolled population was largely nulliparous, the age at delivery was high. Do the authors confirm the value of quartiles (25-59 years)?

[Response]

We appreciate this question. As stated, our hospital is a university hospital and is more likely to be visited by high-risk pregnant women. Furthermore, the average maternal age at first delivery is >31 years in Japan. Since this background might affect the results, we have included the following sentence in the Limitations:

“Fourth, since our hospital is a university hospital that primarily cares for high-risk mothers, there might be a selective bias in the present study.” (p6, lines 204−206)

Reviewer 2 Report

Comments and Suggestions for Authors

Dear Authors,

Very interesting article! I have some comments.

Abstract.

Statistical methods could be omitted so the sentence “The association between GV and insulin secretion, sensitivity, and beta-cell function parameters were analyzed using the Pearson’s test” should be deleted.

Please use the same number of decimals for all p-values, either 3 or 4 (it is your choice, but once you chose, please be consistent). In the Abstract there are p-values with 2, 3, and 4 decimals. This is valid for the entire manuscript; please correct the decimals for all p-values in the article.

2. Materials and Methods.

Please specify what that Pearson’s test is. I suppose it was Pearson’s r correlation coefficient? If so, I would suggest replacing it with Spearman’s rho correlation coefficient due to the fact your variables were not normally distributed.

Please also add a sentence about the descriptive statistics (numbers, proportions, median, IQR). In addition, please add the test you used to check the normality.

3. Results.

Table 1: it is not clear whether you reported median and range (minimum and maximum) or median and IQR (25th and 75th percentiles). That was the reason I asked the descriptive statistics to be specified in Materials and Methods. In addition it is useful to specify this in each table. The p-values in both tables as well as in the text should have the same number of digits.

Table 2. Please specify what you adjusted for.

Figure 1. The figure is very useful however please change the names of the variables to their labels (i.e. “HOMA-R” instead of “HOMA_R” etc.)

I did not see the results of Cochran–Armitage trend analysis?

Author Response

Reviewer 2

Abstract.

Statistical methods could be omitted so the sentence “The association between GV and insulin secretion, sensitivity, and beta-cell function parameters were analyzed using the Pearson’s test” should be deleted.

[Response]

Thank you for your advise. I deleted this sentence.

Please use the same number of decimals for all p-values, either 3 or 4 (it is your choice, but once you chose, please be consistent). In the Abstract there are p-values with 2, 3, and 4 decimals. This is valid for the entire manuscript; please correct the decimals for all p-values in the article.

[Response]

Thank you for your advise. I revised them.

  1. Materials and Methods.

Please specify what that Pearson’s test is. I suppose it was Pearson’s r correlation coefficient? If so, I would suggest replacing it with Spearman’s rho correlation coefficient due to the fact your variables were not normally distributed.

Please also add a sentence about the descriptive statistics (numbers, proportions, median, IQR). In addition, please add the test you used to check the normality.

[Response]

Thank you for your advise. I revised the method.

[Materials and Methods]

“The correlation between GV and insulin parameters was calculated using Spearson’s test.” (p3, lines 101-102)

  1. Results.

Table 1: it is not clear whether you reported median and range (minimum and maximum) or median and IQR (25th and 75th percentiles). That was the reason I asked the descriptive statistics to be specified in Materials and Methods. In addition it is useful to specify this in each table. The p-values in both tables as well as in the text should have the same number of digits.

[Response]

Thank you for your advise. I revised the following sentence.

“Data are presented as the median (range: minimum and max) or n (%).” (p4, lines 122-123)

Table 2. Please specify what you adjusted for.

[Response]

Thank you for your query. I did not adjust this statistical analysis. I performed multiple regression analysis using insulin parameters and GV. Therefore, I revised Table 2.

Figure 1. The figure is very useful however please change the names of the variables to their labels (i.e. “HOMA-R” instead of “HOMA_R” etc.)

[Response]

[Response]

Thank you for your advise. I revised Figure 1.

I did not see the results of Cochran–Armitage trend analysis?

[Response]

As described at Methods section, the trend in the number of abnormal PG during the 75 g-OGTT was evaluated using Cochran–Armitage trend analysis.

Reviewer 3 Report

Comments and Suggestions for Authors

DM is a pathology frequently noticed in daily practice with increasing prevalence worldwide. Over the years many studies assessed the influence of this disease on prognosis, but also its influence on the onset and progression of other conditions. A special group with a considerable risk of the development of DM is represented by pregnant women, but only a few studies focused on the impact and consequence of this pathology in this group of population. The present work may highlight several methods in a better assessment of patients associating GDM, and its results could be considered of great importance as the presence of this disease can be correlated with negative impact not only on the pregnant women, but also on the long-term outcome of the newly borns. The methodology was clearly described and the results were well assessed, consequently the conclusions being in accordance with the findings of this research. One minor comment: as a diet therapy group was also included, for a better understanding, it would be appreciated if more details regrading this particular group should be presented, such as the diet that was used. In addition, for both groups, it would be important to highlighted if the included patients associated or not other pathologies that may influence the long-term prognosis (for both the mothers and newly borns).

Author Response

Reviewer 3

DM is a pathology frequently noticed in daily practice with increasing prevalence worldwide. Over the years many studies assessed the influence of this disease on prognosis, but also its influence on the onset and progression of other conditions. A special group with a considerable risk of the development of DM is represented by pregnant women, but only a few studies focused on the impact and consequence of this pathology in this group of population. The present work may highlight several methods in a better assessment of patients associating GDM, and its results could be considered of great importance as the presence of this disease can be correlated with negative impact not only on the pregnant women, but also on the long-term outcome of the newly borns. The methodology was clearly described and the results were well assessed, consequently the conclusions being in accordance with the findings of this research.

One minor comment: as a diet therapy group was also included, for a better understanding, it would be appreciated if more details regrading this particular group should be presented, such as the diet that was used. In addition, for both groups, it would be important to highlighted if the included patients associated or not other pathologies that may influence the long-term prognosis (for both the mothers and newly borns).

[Response]

We appreciate this feedback. To address your comments, we have added the following sentences:

“All mothers with GDM performed self-blood glucose monitoring before and 2 h after all planned meals and at bedtime in this study. Diet therapy was started immediately after they were diagnosed with GDM. The number of calories for diet therapy was decided based on maternal height and pre-pregnancy body mass index (BMI); height (m)2 × 22 × 30 kcal (pre-pregnancy BMI≥ 25) and height (m)2 × 22 × 30 +350 kcal (pre-pregnancy BMI< 15). When their PG before meal was >90 mg/dL or that at 2 h after meal was 120 mg/dL, the mothers were administered insulin therapy.” (p2, lines 75−80)

“Further studies are required to investigate the usefulness of GV in the management of GDM and reveal the association between maternal and neonatal long-term outcomes and GV.” (p7, lines 210−212)

Round 2

Reviewer 1 Report

Comments and Suggestions for Authors

I don't see the reply to my two major concerns. Again:

1) It is well known that OGTT can have large variability in the same subject. Thus, it is hard to hypotize that a single OGTT may be a good parameter of glucose variability during later gestation.

2) The results show that GV is able to detect differences in insulin parameters. Nevertheless, these differences have no impact in neonatal outcomes. Thus, the significance of this detection seems unclear.

Author Response

Reviewer 1

I don't see the reply to my two major concerns. Again:

  • It is well known that OGTT can have large variability in the same subject. Thus, it is hard to hypotize that a single OGTT may be a good parameter of glucose variability during later gestation.

[Response]

Thank you for your advice. We acknowledge and agree with your concern. However, pregnant women usually receive OGTT only once during their pregnancy. They are subsequently diagnosed with GDM and receive management according to the OGTT result. Although the OGTT results may change, it is necessary to predict the risks or outcomes based on this single occasion. As we described in the Discussion section, managing plasma glucose levels using CGM is not popular in Japan. Therefore, we could not evaluate GV using CGM. We have added the following sentence to the manuscript:

[Discussion]

“Fifth, in the present study, we evaluated GV using the 75 g-OGTT in Japanese mothers with GDM. However, the results of the 75 g-OGTT might change after several tests. Therefore, it might be preferable to use GV measured by CGM when evaluating GV in Japanese mothers with GDM. However, since managing GDM using CGM is not popular in Japan, evaluating GV using CGM was difficult. Further research is required to investigate the usefulness of GV using CGM in Japanese mothers with GDM.” (p6, lines 205–210)

  • The results show that GV is able to detect differences in insulin parameters. Nevertheless, these differences have no impact in neonatal outcomes. Thus, the significance of this detection seems unclear.

[Response]

Thank you for your comment. We understand your concern regarding this issue. However, the incidence of macrosomia and LGA is relatively low among Japanese mothers with GDM since Japanese mothers have a high prevalence of underweight and a low prevalence of obesity. Therefore, we cannot evaluate the association between GV and fetal growth. Although neonatal hypoglycemia could be evaluated in our cases, we could not get the results of plasma glucose levels from medical records from over 60 neonates. Therefore, we did not analyze the association between GV and neonatal outcomes. Nevertheless, if insulin therapy can be predicted during the timing of OGTT, we would be able to strictly manage PG from the timing to diagnose GDM.

If you recommend including details regarding the association between GV and neonatal hypoglycemia, we can provide this. We would like instructions on how to proceed.

Reviewer 2 Report

Comments and Suggestions for Authors

Dear Authors,

Thank you for your efforts to improve the article!

I asked you to choose between 3 and 4 decimals, but you decreased them to 2 decimals. Let me highlight that p<0.05 is not informative at all. Please use 3 decimals (for smaller p use p<0.001) or 4 decimals (for smaller use p<0.0001).

Please add the test for normality that was used.

It seems that the multiple logistic regression you used was not adjusted. Therefore, it is recommended that you adjust for possible confounders like gestational week, BMI, etc.

Excellent study, by the way! I want to see it published.

Author Response

Reviewer 2

Dear Authors,

Thank you for your efforts to improve the article!

I asked you to choose between 3 and 4 decimals, but you decreased them to 2 decimals. Let me highlight that p<0.05 is not informative at all. Please use 3 decimals (for smaller p use p<0.001) or 4 decimals (for smaller use p<0.0001).

[Response]

Thank you for your suggestion. We have revised the p-values accordingly.

Please add the test for normality that was used.

[Response]

Thank you for your advice. We have revised our manuscript by including the following sentence:

[Materials and Methods]

“Since the Shapiro–Wilk test did not show that the data of GV and insulin parameters are consistent distribution, the correlation between GV and insulin parameters was calculated using Spearman’s test.” (p3, lines 101–103)

It seems that the multiple logistic regression you used was not adjusted. Therefore, it is recommended that you adjust for possible confounders like gestational week, BMI, etc.

Excellent study, by the way! I want to see it published.

[Response]

Thank you for your suggestion. We have adjusted the analysis using maternal age at delivery and pre-pregnancy BMI and included the following sentence and Table 2:

[Materials and Methods]

“To investigate the risks of insulin therapy, insulin parameters and GV were evaluated using multiple logistic regression analysis, adjusted for maternal age at delivery and pre-pregnancy BMI.” (p3, lines 103–105)

Round 3

Reviewer 1 Report

Comments and Suggestions for Authors

I recommend including details regarding the association between GV and neonatal hypoglycemia.

Author Response

Reviewer 1

I recommend including details regarding the association between GV and neonatal hypoglycemia.

[Response]

Thank you for your valuable suggestion. We have added the required sentences to our manuscript per your suggestion below. However, as described in the response to revision 2, we could not obtain the results of neonatal PG from the medical records of over 60 neonates. Therefore, we would like you to consider the results of this study as informative.

[Methods]

“Neonatal hypoglycemia is defined as PG <47 mg/dL at 1, 2, or 4 h postpartum [10].” (p3, lines 96–97)

[Results]

“Among the objectives, neonatal hypoglycemia was identified in 77 neonates; however, the results of neonatal PG could not be obtained from the medical records of 66 neonates. The initial increase did not differ between the neonatal hypoglycemia (median: 86 mg/dL, range: 23–157 mg/dL) and non-hypoglycemia (median: 93 mg/dL, range: 24–156 mg/dL) (p=0.13) groups. However, the subsequent decrease was significantly lower in the neonatal hypoglycemia group (median: 12 mg/dL, range: -67 to 117 mg/dL) than in the non-hypoglycemia group (median: 18 mg/dL, range: -63 to 108 mg/dL) (p<0.05).” (p5, lines 147–153)

[Discussions]

“Although the subsequent decrease was significantly lower in the neonatal hypoglycemia group than in the non-hypoglycemia group in the present study, we could not obtain the results of neonatal PG from the medical records of over 60 neonates. Therefore, studies with larger sample sizes are needed to evaluate the association between GV and neonatal hypoglycemia.” (p7, lines 211–215)

Reviewer 2 Report

Comments and Suggestions for Authors

Thank you! The manuscript is improved. 

Author Response

Thank you for your help!